# OpenReview forum: "Hyperbolic Variational Autoencoders for Phylogenetic Latent Spaces: Geometric Priors for Evolutionary Sequence Modeling"
_mathai.club/MathAI/2026/Conference — Submitted to 2026_

### Official Review · Reviewer_8jBX · 2026-03-11
**Accept (on the committee discretion).**

**Rating:** 8
**Confidence:** 3

**Review:**

General recommendation: Accept.
This submission fits MathAI scope. It introduces PhyloVAE: hyperbolic VAE in Poincaré ball for sequence data. Implementation to phylogenetic analysis is shown very common. Problem of phylogeny are not discussed, and the data are not shown and not referenced - Like PFAM database for proteins, data on COVID-19. Thus, the work is superficial for biology section of the conference.
The abbreviaions like VAE, ELBO are known, but never shown or referenced.
Mathematically the terminology is correct, and formalism is presented.
1. Mathematical Rigor: high.
Theorems: ELBO decomposition, distortion bounds (Papadopoulos), posterior concentration (empirical processes), info efficiency.  Solid proofs.
2. Novelty & Contribution: high.
First hyperbolic VAE for phylogenetics; hyperbolic prior matches tree geometry.
3. Relevance to MathAI: high.
Appropriate Tracks A/B/D: Riemannian manifolds, variational inference, hyperbolic embedding theorems for bio-AI phylogenetics.
4. Technical Quality: good.
Correct hyperbolic ops, Riemannian Adam. Benchmarks (Pfam/GISAID/RNA) validate (but no reference, no explanation).
5. Clarity & Presentation: good.
Clear structure, figs (t-SNE, distortion), tables. Precise notation.
6. AI-Generation Risk: low
Sophisticated theory/experiments; human-level proofs/insights.
Overall Recommendation
Accept (Track B/D oral). Landmark geometric deep learning for phylogenetics; broad impact. As publication is not yet ready. The references are rather old. No references to biological databases.

---

### Official Review · Reviewer_vcuX · 2026-03-11
**Reject: an interesting high-level idea undermined by mathematically invalid theory, non-auditable experiments, and severe presentation/referencing failures.**

**Rating:** 2
**Confidence:** 5

**Review:**

# Summary

This paper proposes **PhyloVAE**, a hyperbolic variational autoencoder for biological sequences, motivated by the intuition that phylogenetic structure is tree-like and therefore better matched by negatively curved latent spaces than by Euclidean ones. The motivation is reasonable at a high level, and the application area is important. Unfortunately, the submission is **not scientifically reliable in its current form**.

My overall assessment is **strong reject**.

The central theoretical claims appear incorrect or, at best, unproven; the empirical section is not sufficiently specified to be reproducible or trustworthy; multiple figures are mismatched with their captions; several references are clearly incorrect or unrelated; and the manuscript repeatedly makes claims about **phylogenetic inference** without actually defining a generative model over phylogenetic trees. As written, I do not find the paper credible.

---

# Evaluation

## Quality: **Very weak**

The paper’s most important weakness is that the theory is not just incomplete, but appears fundamentally flawed.

### 1. The hyperbolic KL derivation is not credible

The paper claims an **analytic hyperbolic KL divergence** between wrapped normal distributions and presents Eq. (12) as if it were essentially the Euclidean Gaussian KL in tangent coordinates. That is not justified.

For a wrapped normal defined by
\[
z = \exp_\mu(\epsilon), \qquad \epsilon \sim \mathcal{N}(0,\Sigma),
\]
the density on the manifold should involve a change-of-variables term of the form
\[
\log q(z)
=
\log \mathcal{N}(\epsilon;0,\Sigma)
-
\log \left| \det D\exp_\mu(\epsilon) \right|,
\]
or equivalently a Jacobian/volume correction in log-map coordinates. The manuscript’s “proof” replaces the manifold density with a tangent-space Gaussian plus a first-order expansion, but then states an **exact closed-form KL**. This is a serious gap. A first-order normal-coordinate approximation does **not** establish the exact formula claimed.

In other words, the paper jumps from an approximation to an equality. That is not acceptable for a theorem-level claim.

### 2. The distortion theorem is conceptually broken

Theorem 2 is presented as a major theoretical result, but it is deeply problematic.

The paper states that for a phylogenetic tree with hyperbolicity constant \(\delta\), one gets distortion \(O(\delta)\), and even claims:
- balanced trees have \(\delta = O(\log n)\),
- star trees have \(\delta = O(n)\).

This is a fundamental error. **Tree metrics are \(0\)-hyperbolic.** In standard Gromov hyperbolicity,
\[
\delta_{\text{tree}} = 0
\]
for any tree metric, including balanced trees and star trees.

That single mistake already severely undermines the theoretical section.

Worse, the theorem assumes
\[
d_{\mathbb{H}}(z_i,z_j)=\lambda d_T(i,j)+\epsilon_{ij},
\]
i.e. it essentially assumes the embedding already preserves tree distances up to controlled error. The conclusion then says the distortion is \(O(\delta+\rho)\). This is close to circular: the statement largely assumes what it claims to prove.

### 3. The posterior concentration result is not meaningful as stated

Theorem 3 claims posterior concentration around the **MLE phylogeny** at rate \(O(n^{-1/2})\) in Wasserstein distance on the “Phylogenetic Orangespace.” This section is not rigorous enough to evaluate and, frankly, reads as non-mathematical.

Problems include:

- The model does **not** explicitly parameterize a phylogenetic tree \(T\), branch lengths, or a substitution process.
- A latent vector per sequence is **not** the same thing as a phylogeny.
- The object called “Phylogenetic Orangespace” is undefined and appears to be a typo or invented term.
- The notation
  \[
  W_{\mathrm{ot}}(\mathrm{Law}(\mu_\phi(X)), \theta^*)
  \]
  mixes a law/distribution with a parameter point in a way that is not properly defined.
- “The ELBO concentrates around the true log-likelihood by standard results” is far too vague for such a strong claim.

This theorem is not close to publication quality.

### 4. The mutual information theorem is hand-wavy and unsupported

Theorem 4 appears to be an informal intuition written in theorem format. There is no rigorous argument connecting phylogenetic pairwise distances, bit complexity, Euclidean embedding distortion, and a lower bound on \(I(X;Z)\) in the form claimed. The dimensional scaling claim is also unconvincing. This section should not be presented as a theorem.

---

## Clarity: **Poor**

The manuscript is difficult to trust even at the presentation level.

### 1. Key terms and objects are undefined or misused

- The wrapped normal notation is sloppy; the definition includes an \(x\) argument that is not used as a density argument.
- The paper speaks about “posterior concentration around the MLE tree” even though no tree-valued latent variable is modeled.
- “Phylogenetic Orangespace” is undefined.
- The decoder is described only superficially, while the appendix allegedly contains details that do not actually appear.

### 2. Figures are seriously mismatched with their captions

This is one of the strongest empirical red flags in the paper.

- **Figure 4** is captioned as a comparison of distance distortion heatmaps for balanced and star-shaped trees, but the displayed figure is a **line plot of reconstruction accuracy versus latent dimension**.
- **Figure 5** is captioned as showing reconstruction examples and alignments, but the displayed figure is a **generic bar chart / ablation-style plot**.
- **Figure 3** claims “clear phylogenetic clustering in hyperbolic space,” but the shown hyperbolic panel appears visually uninformative, with points collapsed near the center.

These are not minor copy-editing issues. They make the empirical section non-auditable.

### 3. The paper contains obvious citation and formatting problems

Multiple references are garbled, duplicated, or unrelated to the claims they supposedly support. This substantially reduces confidence that the authors understand the relevant literature.

---

## Originality: **Potentially interesting idea, but weakly substantiated**

The high-level intuition—using hyperbolic geometry for tree-like biological structure—is reasonable and potentially interesting. However, the novelty is overstated.

There is already substantial prior work on:
- hyperbolic embeddings,
- hyperbolic neural networks,
- latent-variable models in non-Euclidean spaces,
- and machine learning for phylogenetic or hierarchical structure.

So the real novelty would need to come from one of the following:
1. a genuinely new and correct hyperbolic VAE derivation,
2. a biologically meaningful phylogenetic generative model,
3. strong theoretical guarantees that are both correct and relevant,
4. or compelling experimental evidence on carefully specified tasks.

At present, the paper does not convincingly deliver any of these.

---

## Significance: **Currently low**

If the work were correct, the topic could be significant. But significance depends on credibility. Right now, the theory is not reliable and the experiments are not reproducible enough to support the claims. Therefore the current significance is low.

---

# Pros

- The paper tackles a meaningful problem: biological sequence variation is indeed highly structured and often tree-like.
- The geometric motivation for considering hyperbolic latent spaces is intuitively sensible.
- The manuscript attempts to combine theory, algorithms, and experiments, which is the right ambition for this area.
- The empirical tables suggest the authors are aiming to evaluate both reconstruction and structure preservation, which is directionally appropriate.

---

# Cons

## Major cons

1. **Core theory is not credible.**
   The KL derivation ignores manifold Jacobian terms while claiming an exact analytic result.

2. **The hyperbolicity discussion is mathematically wrong.**
   Any tree metric is \(0\)-hyperbolic, so the paper’s discussion of \(\delta\) for balanced and star trees is fundamentally incorrect.

3. **The main distortion theorem is close to tautological.**
   It assumes approximate preservation of phylogenetic distances and then concludes bounded distortion.

4. **The paper does not actually model phylogeny.**
   There is no explicit latent tree, no branch lengths, no substitution model, no ancestral reconstruction, and no probabilistic tree likelihood.

5. **The posterior concentration claim is unsupported and conceptually incoherent.**

6. **The empirical section is not reproducible.**
   Missing details include preprocessing, train/validation/test splits, sequence lengths, alignment handling, tree-construction protocol, baseline configurations, seed counts, and statistical testing.

7. **The evaluation metric is mishandled.**
   The paper writes BLEU incorrectly. Standard BLEU is
   \[
   BLEU = BP \cdot \exp\left(\frac{1}{4}\sum_{n=1}^4 \log p_n\right),
   \]
   not an arithmetic average of \(n\)-gram precisions.

8. **Figures are mismatched with captions.**
   This is a severe reliability issue.

9. **References are broken or unrelated.**
   Several citations appear incorrect enough to cast doubt on the entire scholarship of the paper.

10. **Claims of being the “first” or of providing rigorous guarantees are not adequately supported.**

## Minor cons

- Notation is inconsistent and occasionally incorrect.
- Appendix promises details that are absent.
- Placeholder text remains (e.g., institution acknowledgments).
- Several sentences are overstated and read more like marketing than scientific argument.
- The paper repeatedly uses proof sketches where full proofs are absolutely necessary.

---

# Additional comments

The biological connection is also shallower than the paper suggests. Sequence evolution on a phylogeny is not just “hierarchical structure”; it is governed by explicit stochastic processes on trees. A geometry-aware latent space may be useful as an inductive bias, but that is **not** equivalent to doing phylogenetic inference. The paper conflates these two things throughout.

I would have been much more positive about a modest paper that said:

> “We study whether hyperbolic latent spaces improve representation quality for sequence families with tree-like structure.”

That would be a reasonable empirical representation-learning paper. But this submission instead makes very strong claims about exact KL formulas, embedding theorems, posterior concentration around phylogenies, and rigorous biological significance, without supplying the required mathematical or empirical support.

---

# Recommendation

**Strong Reject**

The paper has an interesting motivation, but the current submission is not ready for publication. To become viable, it would require:

1. a complete rewrite of the theory with correct manifold probability calculations,
2. removal or replacement of false theorems,
3. an explicit and biologically meaningful phylogenetic probabilistic model,
4. fully reproducible experiments,
5. corrected figures and captions,
6. and a careful reconstruction of the related-work section and bibliography.

At present, I do not trust the mathematical claims, I do not trust the empirical claims, and I do not trust the references enough to recommend acceptance.

---

### Official Review · Reviewer_5rHw · 2026-03-12
**Promising geometric motivation, but the paper does not substantiate its theoretical claims**

**Rating:** 5
**Confidence:** 4

**Review:**

**Summary**

This paper proposes PhyloVAE, a hyperbolic variational autoencoder for biological sequence data, motivated by the tree-like nature of phylogenetic structure. The high-level idea is reasonable: hyperbolic latent geometry is a plausible inductive bias for hierarchical data. However, the current paper does not provide sufficiently reliable support for its main claims.

**Strengths**

The motivation is clear, the topic is relevant, and the paper aims to connect geometry-aware latent modeling with biological sequence analysis. The overall direction is interesting.

**Weaknesses**

- **Theoretical claims are stronger than what is actually shown.** The paper states in S1 that Section 4 contains ``formal proofs'', but the key results are presented only with proof sketches. This is especially problematic for Theorem1 (S3.3) and Theorems2--4 (S4), where the paper moves from approximations or informal arguments to theorem-level conclusions without enough justification.

- **The paper makes strong phylogenetic claims without defining the relevant objects precisely enough.** In particular, Theorem3 (S4.2) claims posterior concentration around the MLE phylogeny, but the paper does not clearly specify the mathematical relationship between the latent representation and a tree-valued phylogenetic object. As written, this claim is too strong for the level of formalization provided.

- **Model description is not sufficiently reproducible.** The decoder description in S3.4 is very brief, and the paper says that attention-based architectural details are given in the Appendix, but the Appendix only lists hyperparameters, complexity, and numerical stability notes.

- **There are serious presentation inconsistencies.** Figures4 and 5 do not match their captions. This is not a minor editorial issue: it directly undermines confidence in the empirical section and makes the results difficult to verify from the manuscript itself.

---

### Decision · Program_Chairs · 2026-03-14

**Decision:**

Reject

**Comment:**

After careful evaluation by the Program Committee, we regret to inform you that your submission has not been accepted for presentation at MathAI 2026.

All submissions underwent a rigorous two-stage review process. Unfortunately, the reviewers identified one or more of the following concerns with your paper:

- Insufficient mathematical rigor or novelty relative to the existing body of work in the field;
- Presentation of results that substantially overlap with or rephrase previously published findings without clear original contribution;
- Significant issues with technical quality, including but not limited to broken or non-existent references, unsupported claims, or methodological gaps;
- Indications that the manuscript may have been generated with the assistance of large language models without substantial original intellectual contribution by the authors.

We received a large number of submissions this year, and the selection process was highly competitive. We encourage you to carefully consider the reviewers’ feedback (available through OpenReview), revise your work accordingly, and consider submitting an improved version to a future edition of MathAI or to another appropriate venue.

We appreciate your interest in MathAI and hope you will continue to engage with the conference community.

With kind regards,

MathAI 2026 Program Committee
International Conference on Mathematics of Artificial Intelligence
https://mathai.club
OpenReview: https://openreview.net/group?id=mathai.club/MathAI/2026/Conference
Telegram: https://t.me/MathAI_club
Email: mathai.club@yandex.ru